# Label-Free Mass Spectrometry-Based Quantification of Linker Histone H1 Variants in Clinical Samples

**DOI:** 10.3390/ijms21197330

**Published:** 2020-10-04

**Authors:** Roberta Noberini, Cristina Morales Torres, Evelyn Oliva Savoia, Stefania Brandini, Maria Giovanna Jodice, Giovanni Bertalot, Giuseppina Bonizzi, Maria Capra, Giuseppe Diaferia, Paola Scaffidi, Tiziana Bonaldi

**Affiliations:** 1Department of Experimental Oncology, IEO, European Institute of Oncology IRCCS, 20139 Milan, Italy; EvelynOliva.Savoia@ieo.it (E.O.S.); stefania.brandini@ieo.it (S.B.); giovanna.jodice@ieo.it (M.G.J.); giovanni.bertalot@ieo.it (G.B.); giuseppe.diaferia@ieo.it (G.D.); 2Cancer Epigenetics Laboratory, Francis Crick Institute, London NW1 1AT, UK; cristina.morales@crick.ac.uk (C.M.T.); Paola.Scaffidi@crick.ac.uk (P.S.); 3Biobank for Translational Medicine Unit (B4MED), Department of Pathology and Laboratory Medicine, IEO, European Institute of Oncology IRCCS, 20141 Milan, Italy; giuseppina.bonizzi@ieo.it (G.B.); maria.capra@ieo.it (M.C.)

**Keywords:** epigenetics, label-free quantification, linker histone H1, mass spectrometry, proteomics, laser micro-dissection, breast cancer

## Abstract

Epigenetic aberrations have been recognized as important contributors to cancer onset and development, and increasing evidence suggests that linker histone H1 variants may serve as biomarkers useful for patient stratification, as well as play an important role as drivers in cancer. Although traditionally histone H1 levels have been studied using antibody-based methods and RNA expression, these approaches suffer from limitations. Mass spectrometry (MS)-based proteomics represents the ideal tool to accurately quantify relative changes in protein abundance within complex samples. In this study, we used a label-free quantification approach to simultaneously analyze all somatic histone H1 variants in clinical samples and verified its applicability to laser micro-dissected tissue areas containing as low as 1000 cells. We then applied it to breast cancer patient samples, identifying differences in linker histone variants patters in primary triple-negative breast tumors with and without relapse after chemotherapy. This study highlights how label-free quantitation by MS is a valuable option to accurately quantitate histone H1 levels in different types of clinical samples, including very low-abundance patient tissues.

## 1. Introduction

In the eukaryotic nucleus, genomic DNA is packed into chromatin, which regulates all nuclear processes involving DNA, including DNA replication, DNA repair and transcription. Histones represent the major protein component of chromatin and comprise core and linker histones. Core histones (H2A, H2B, H3 and H4) assemble in a histone octamer, around which approximately 146 bp of DNA is wrapped, forming a nucleosome. Linker histone H1 binds the free DNA (~20 bp-long) present between individual nucleosomes, contributing to the formation of higher-order chromatin structures. Together, the nucleosome, linker DNA and linker histone H1 form the chromatosome. Core histones exert their function mainly through a number of reversible post-translational modifications (PTMs) that can be deposed at their *N*-terminal tails, whose role in regulating gene expression has been extensively studied and elucidated in recent years [1]. In contrast, much less is known regarding the role of histone H1 besides its general chromatin condensation function, although increasing evidence indicates histone H1 potential to regulate transcription in a locus-specific manner [2,3,4].

Linker histones comprise 11 variants (or subtypes) in human and mouse, including seven somatic variants (H1.0, H1x and H1.1 to H1.5), three testis-specific variants (H1t, H1T2 and HILS1) and one oocyte-specific variant (H1oo). Histones H1.1 to H1.5 are expressed in a replication-dependent manner, while histones H1.0 and H1x are replication-independent variants transcribed throughout the cell cycle [5]. The definition of the structures of the chromatosome core particle and of the 30 nm nucleosome fiber revealed that histone H1 variants can bind differently to the nucleosome, determining distinct higher-order chromatin structures and contributing to the regulation of nuclear functions, including transcription, DNA replication and repair, and genome stability (reviewed in [3]). In addition to existing as different variants, histone H1 contains different PTMs that may modulate its functions, which include methylation, phosphorylation, acetylation, ubiquitination, citrullination, formylation and ADP ribosylation [6].

Alterations in the levels of bulk histone H1 and of specific variants have been observed in cancer (reviewed in [7]). For instance, a 40% reduction of bulk H1 mRNA levels was found in ovarian adenocarcinoma [8], and a number of variant-specific changes have been observed in different types of tumors [7]. Generally, replication-dependent variants are upregulated in tumors compared with normal tissues, as the result of increased cell proliferation. Histone H1.0, which is usually present at high levels in normal tissues, shows heterogeneous patterns in cancers and is overall reduced, especially in aggressive and undifferentiated tumors [7]. These results suggest that H1 levels could be useful biomarkers to discriminate benign and malignant lesions and to possibly provide information about patient prognosis.

Finally, recurrent mutations of histone H1 subtypes have been found in different types of cancers [7]. Although their relevance in tumorigenesis is currently poorly understood, this finding could support a driver role played by linker histones in cancer, whose investigation could not only lead to the identification of biomarkers for patient stratification, but possibly also to the discovery of novel epigenetic mechanisms and therapeutic strategies.

Although histone H1 investigation in clinical samples relied so far mostly on the analysis of RNA expression levels or antibody–based detection of protein levels, none of these approaches is ideal. First, alterations in mRNA levels may not correspond to changes in protein abundance, since H1 genes are regulated at the post-transcriptional level and subjected to translational control [9]; second, antibody-based methods, such as immunohistochemistry, require variant-specific antibodies, which represents a major challenge given the high similarity of the H1.1–H1.5 subtypes (Figure 1). In this context, mass spectrometry (MS)–based proteomics offers the ideal tool to analyze in a quantitative manner the protein levels of different histone H1 variants, also in clinical samples.

In this study, we applied a label-free MS-based proteomics approach to the analysis of histone H1 variants in clinical samples, including very low abundance tissues. After setting up the method in cell lines and laser micro-dissected mouse samples, we applied it to human breast cancer samples, detecting differences in several variants between triple-negative breast cancers with different outcomes.

## 2. Results

### 2.1. Quantification of Histone H1 Variants through the MaxLFQ Algorithm

Linker histones represent the most divergent histone protein family. Several positively charged residues in histone H1 globular domains that are important for nucleosome binding are conserved among species and individual variants [10], but the *N*– and *C*–terminal tails are much less conserved among the variants, thus allowing them to be distinguished by MS. Figure 1A shows the alignment of mouse somatic histone H1 variants, where the unique amino acid sequences detected in our experiments are highlighted. Either lysates or nuclear extracts were loaded on a polyacrylamide gel, and a large band (20–45 kDa) around the size of histone H1 was excised for in-gel digestion with trypsin (Figure 1B). The aim of digesting only proteins corresponding to a specific molecular weight range was to enrich histone H1 relative to the rest of the proteins, while at the same time retaining a sufficient number of background proteins to allow an adequate normalization of protein amounts within the sample. Furthermore, gel separation removes MS contaminants that are often present in the preparations obtained from clinical samples (e.g., detergents and tissue-embedding compounds).

To quantify histone H1 variants after MS acquisition, we took advantage of a label-free quantification (LFQ) algorithm, named MaxLFQ [11], which relies on the extraction of ion intensities and is implemented in the freely available MaxQuant computational proteomics platform [12]. As opposed to labelling strategies, label-free quantification approaches are straightforward and cost-effective, and they are applicable to any number and any types of samples, including patient-derived specimens.

As a case study to test our approach, we processed mouse embryonic fibroblasts (MEFs) expressing a doxycycline-inducible H1.0-targeting shRNA, which were left untreated (WT) or treated with doxycycline for 10 days, inducing histone H1.0 knock-down (H1.0 KD). We processed biological triplicates for whole lysates and nuclear extracts, obtaining very similar results with the two sample preparation methods (Figure 1C–E). In total, we identified 40 unique peptides deriving from the digestion of histone H1 variants (Figure 1C). Histone H1.1 was identified with the highest number of unique peptides (ten), while histone H1.3 with only one peptide. Most of the peptides were identified by MS/MS, while some were identified through the MaxQuant “match between runs” option, which allows transferring identifications of MS1 features among different samples based on mass and retention time values. For some of the peptides (e.g., histone H1.1 peptides), the identification by MS/MS was slightly more efficient from the nuclei preparation. Accordingly, the relative abundance of linker histone variants compared with the other proteins present in the samples (as assessed by using the intensity-based absolute quantification (IBAQ) score, see Experimental Procedures) appears to be slightly higher from in-gel digested nuclei compared with whole lysates, although their relative proportion remains the same (Figure 1D). As expected, we detected a significant decrease of histone H1.0 in H1.0 KD cells processed with either of the two in-gel digestion methods. This decrease was paralleled by an increase of all the other H1 isoforms, with the exception of histone H1x (Figure 1E), a compensation that was previously reported [13]. Taken together, these results show that the two in-gel processing methods perform similarly and can both be employed. Because isolating nuclei appears to provide slightly better results, this approach is preferable, when feasible, and was used for the next experiments in cells. However, processing whole lysates may be preferred to avoid loss of material.

Next, we compared the results obtained by MS with two alternative methods commonly employed to study histone H1 variants, namely immunofluorescence and gene expression analysis. For this analysis, we focused on the H1.0 variant, for which reliable specific antibodies are available. We quantified histone H1.0 protein levels in MEF cells WT, H1.0 KD, and treated with the histone deacetylates inhibitor (HDACi) Quisinostat, which has been shown to cause an increase of histone H1.0 level [14], obtaining remarkably similar results by MS (Figure 2A, left panel) and immunofluorescence (Figure 2A, middle panel, and Figure 2B). The levels of mRNA were also similar (Figure 2A, right panel), although displaying a more pronounced increase of histone H1.0 after treatment with the HDAC inhibitor (a 25-fold change compared with 8-fold change by MS). This discrepancy shows that mRNA levels are not entirely representative of the protein levels, likely due to post-transcriptional regulation, and emphasizes the need for an accurate method of protein quantification.

Finally, to test the accuracy of the MS quantification, we mixed H1.0 KD and HDAC-treated cells in different proportions and compared the observed and expected histone H1 levels (Figure 2C and Appendix A). The correlation between observed and expected levels (normalized over the KD sample) showed very good linearity for both histone H1.0 (Figure 2C, left panel) and the other histone H1 variants that displayed a change between HDACi-treated cells and H1.0 KD cells (Figure 2C, right panel).

### 2.2. MS-Based LFQ Quantification of Histone H1 Variants in Low-Abundance Samples

With the goal of developing an approach suitable also for low sample amounts, we performed serial dilutions (starting from 10 µg of loaded proteins) of the digested peptides obtained from nuclear extracts (Figure 3A). This experiment tests two important aspects of the performance of the MaxLFQ algorithm: (1) the ability to correct for different amounts of input of material and (2) the ability to quantify histone H1 variants from low-abundance samples. All the somatic histone isoforms were identified and quantified in all the dilutions, with the exception of histone H1x, which appears to be the least abundant isoform in these samples (see Figure 1D) and was not detectable in the 1:125 dilution (Figure 3B). Although the number of peptides unique for specific histone H1 variants decreases with decreasing amounts of injected material, as expected, most peptides could still be identified, either by MS/MS or with the help of the “match between runs features”, from the 1:25 dilutions (Appendix A). The correlation of the LFQ values showed extremely high Pearson’s correlation scores (>0.95) for all the dilutions, with the exception of the 1:125, where the values were more variable (Figure 3C). Furthermore, the decrease of H1.0 in H1.0 KD cells was detectable in all the samples, while the increase of the other isoforms was detectable in all dilutions except for the 1:125 (Figure 3D). These results indicate that the MaxLFQ algorithm performs very well in normalizing similar samples available in very different amounts, and that it can accurately quantify histone H1 changes from as low as a few ng of injected peptide mixture.

### 2.3. Quantification of Histone H1 Variants from Laser Micro-Dissected Samples

Although serial dilutions of digested peptides can provide an indication on the amount of material needed, it does not mimic the real experimental procedure of processing low amounts of cells/tissues, which involves relevant loss of material throughout the procedure. Therefore, we tested our sample preparation approach in combination with the MaxLFQ algorithm on laser micro-dissected samples containing from approximately 1000 to 20,000 mouse pancreas cells. We digested whole-tissue lysates to minimize loss of material, and we processed also whole-tissue sections to be used as a reference. To account for possible differences due to storage, we used samples stored as formalin-fixed paraffin-embedded (FFPE) or optimal cutting temperature (OCT)–frozen tissues. Formalin fixation followed by paraffin embedding is the most common tissue preservation technique used in hospitals for the management of patient samples, but it can cause the appearance of artefactual modifications that can interfere with proteomics analyses and is usually associated with lower yields of identified/quantified proteins [15]. OCT–embedding is another storage method used in tissue biobanks. Typically, frozen tissues are considered of higher quality compared with FFPE, but the OCT compound represents a strong contaminant that must be carefully removed prior to MS analysis. Remarkably, all somatic H1 variants were identified from all the cell amounts for OCT samples and most of them also for FFPE samples (Figure 4A). Fewer unique peptides could be identified from LMD samples compared with whole sections, and in some instances (e.g., H1.1 peptides) identification matching from whole sections was indispensable to identify a specific histone H1 variant, strongly suggesting the need for an abundant reference sample to be run in parallel and to be used as a source of MS/MS for the transfer of identifications to less abundant samples. Most histone H1 variants could be quantified through the MaxLFQ algorithm from all the samples, with the exception of histone H1.1 and H1x, which were not quantified in a few of the samples containing the lowest numbers of cells (Figure 4B). This result is likely due to the particularly low abundance (<0.15%) of these two variants in mouse pancreas (Figure 4B). The correlation of the MaxLFQ values among laser micro-dissected samples was remarkably high (>0.94), even for the samples derived from the lowest number of cells (Figure 4C), indicating that this workflow has the potential to be applied to very small tissue areas.

Of note, the pattern of linker histone variants in whole sections was slightly different from that observed in laser micro-dissected acini, likely as a consequence of the higher heterogeneity found in whole-tissue sections (Appendix A). Importantly, our workflow is applicable to both OCT and FFPE samples, which performed very similarly (Figure 4 and Appendix A). The almost perfect correlation of MaxLFQ values obtained from FFPE and OCT samples also suggests that potential artefacts due to FFPE storage do not influence the quantification of histone H1 isoforms (Appendix A).

### 2.4. Quantification of Histone H1 Variants from Patient Breast Cancer Tissue

Next, we applied our methods to patient-derived breast cancer samples in two distinct settings. Because of sequence differences among species, some of the peptides detected from human histone H1 variants are different from the mouse ones (Appendix A). As a proof-of-concept for the applicability of label-free MS quantitation of histone H1 variants to laser micro-dissected samples, we compared normal and infiltrating carcinoma areas, which were laser micro-dissected from the same patient specimen (Figure 5A). We collected four normal and four tumor areas containing approximately 1800–4500 cells (corresponding to an area of 0.26–0.68 mm^2^). Despite the variability found within areas collected from the same tissue regions, which may reflect the reported heterogeneity in histone H1 expression [16], we detected a decrease/decreasing trend of histone H1.0, which has been reported to be generally reduced in tumors [7], while the other variants did not show any significant changes (Figure 5B).

We also analyzed the levels of histone H1 isoforms in the context of tumor recurrence by profiling 37 breast cancer samples belonging to the triple-negative subtype, of which 19 with and 18 without relapse three years after treatment, matched for clinical features and adjuvant chemotherapy. The samples were stored in OCT, and whole slices were processed (Appendix A). Despite some variability among samples, which was expected given that the triple-negative breast cancer subtype comprises a very heterogeneous group of tumors, we found a significant decrease in the levels of all somatic histone variants in relapsing tumors (Figure 5C–E). Most of these changes were confirmed in an independent dataset composed of nine FFPE samples (five without and four with relapse), for which whole proteomes were acquired (Figure 5D–F). These results may represent potential novel predictive biomarkers for the stratification of triple-negative breast cancer patients and provide the starting point for further investigations into the epigenetic mechanisms leading to tumor recurrence.

## 3. Discussion

In this study, we show the feasibility of applying a simple label-free MS-based approach for the quantitative and simultaneous analysis of histone H1 variant levels from clinical samples. This method takes advantage of the MaxLFQ algorithm, which has been extensively used in quantitative global proteomics studies, but whose application to protein isoforms, such as histone H1 variants, has not been investigated so far. Our experiments show that the MaxLFQ algorithm allows a precise quantification of histone H1 isoforms even in the presence of different amounts of starting material and from very low-abundance samples. Although the MaxLFQ quantification of histone H1 isoforms can be used from unfractionated samples (data not shown), we chose to perform a SDS–PAGE separation combined with an in-gel digestion prior to MS analysis, in order to eliminate MS contaminants from the sample preparation and to select a specific MW range for digestion, which enriches histone H1 relative to background proteins. In addition, this in-gel strategy allows performing histone H1 and core histone PTM analyses, which we typically carry out through in-gel digestion followed by protein derivatization ([17] and unpublished data), from the same samples. This allows a comprehensive epigenetic profiling of samples that may be available in very limited amounts, such as clinical specimens. In this regard, we also verified that our method allows quantifying potentially all histone H1 variants from as low as 1000 laser micro-dissected cells, although some variants may be missed if expressed at particularly low levels in specific tissues.

To aid the identification of histone H1 peptides from samples available in low amounts, we took advantage of the “match between runs” feature available in MaxQuant, which was used to transfer identifications, derived from a standard sequence database searching approach, from an abundant reference sample to low-abundance samples. An alternative approach is represented by spectral library searching, where observed MS/MS spectra are searched against a library of experimental MS/MS spectra to assign identifications [18]. Because the library spectra contain peaks from non–canonical fragment ions that may not be present in a standard database containing in silico digested proteins, as well as intensity information, spectral library searching is particularly useful to increase identification rates for lower-quality spectra, such as those resulting from low-abundance samples. Histone H1 variants could also be analyzed though targeted MS approaches [19], where proteotypic peptides, namely peptides that uniquely represent a target proteins/protein variant and that can be detected in a reproducible manner, can be followed with increased throughput and sensitivity. Our results could help the design of targeted strategies by providing information on histone H1 peptides that can be reliably detected across the samples analyzed in this study.

We verified that histone H1 variants can be analyzed from frozen (including OCT frozen) as well as FFPE tissues, thus including essentially all types of patient samples available from hospital biobanks. For the proof-of-concept laser micro-dissection experiments performed on patient tissues, we isolated different tumor/normal regions to be analyzed based on morphological evaluation of the tissue. As an alternative, the choice of the tissue areas could be based on the spatial molecular profiles obtained by matrix-assisted laser desorption/ionization (MALDI) imaging. This technique combines MS ability to analyze in a comprehensive and unbiased manner a high number of analytes, such as protein or peptides, with the capability of obtaining a spatial distribution of such analytes within tissue sections [20]. MALDI imaging has emerged as a useful tool for cancer diagnosis and prognosis, determination of tumor margins and investigation of tumor heterogeneity [21]. In particular, it has been employed to guide the definition of different areas within heterogeneous breast cancers to be analyzed by microproteomics [22]. Similarly, one can imagine that MALDI imaging profiling of proteins, or ideally histone H1 variants, could be used to select areas to be laser micro-dissected and processed for MS analysis. A MALDI imaging workflow was developed and used to characterize the in situ distribution of all somatic H1 subtypes (except H1.1) in the mammalian brain [23]. This method has also been applied in preliminary studies to cancer patient samples [24].

The analysis of histone H1 variants in breast cancer clinical samples revealed a significant decrease of various variants in triple-negative breast tumors with worse outcome. Usually, H1.0 is highly expressed in normal tissues and is downregulated in various types of cancers, as well as in higher-grade and more aggressive tumors [25,26]. Furthermore, *H1F0* is an independent predictor of patient survival in breast cancer, especially in triple-negative breast cancer [16]. Consistent with these observations, we found a decrease of H1.0 in tumor compared with normal cells, and in tumors showing relapse. Among the breast cancer molecular subtypes, the triple-negative group includes tumors without any known molecular targets, whose treatment is mostly limited to chemotherapy and is often not successful. The recent finding that high H1.0 levels can be restored by Quisinostat suggests that our method could be useful to stratify patients that would most benefit from Quisinostat treatment to prevent disease relapse [14]. Interestingly, we found that the other histone H1 variants were also decreased in triple-negative tumors showing relapse in both the datasets analyzed. Although cell-cycle dependent isoforms are often found at increased levels in tumors with higher proliferation rates and more aggressive features, alterations of histone H1 patters can be context-specific. For instance, it has been shown that H1.1, H1.4 and H1x, in addition to H1.0, were significantly reduced in malignant ovarian adenocarcinomas compared with benign adenomas [8]. Our results suggest that epigenetic mechanisms may be involved in the process of tumor relapse, and that the change in histone H1 variants will be worth further investigations both as potential biomarkers predictive of patient outcome and as an epigenetic mechanism underlying tumor recurrence. It will also be interesting to investigate histone H1 at the level of PTMs associated with different variants, which can be performed from the same data acquired to quantify histone H1 amounts, as our protocols preserve modifications. This information, together with the analysis of core histone PTMs, will contribute to gaining a more complete picture of epigenetic mechanisms linked with cancer.

## 4. Materials and Methods

### 4.1. Tissue Culture

MEFs expressing a doxycycline-inducible H1.0-targeting shRNA (5′UAGCAAAUUCGAAUCAACUGGA-3′, Mirimus, Brooklyn, NY, USA) were cultured in Dulbecco’s Modified Eagle’s Medium (DMEM), supplemented with 10% FBS, 2 mM L–glutamine, 100 U/mL penicillin and 100 µg/mL streptomycin at 37 °C in 5% CO_2_. To induce H1.0 knockdown, MEFs were treated with 1 μg/mL of doxycycline (Merck KGaA, Darmstadt, Germany, D9891) for 10 days (H1.0 KD). To increase H1.0 levels, MEFs were treated with 100 nM Quisinostat (Insight Biotechnology, Wembley, UK, HY–15433-1mL) for 24 h. Untreated MEFs were used as control.

### 4.2. Patient Samples

The use of patient samples was conducted in accordance with the Declaration of Helsinki. Patient samples stored in OCT were obtained from the Biobank for Translational Medicine Unit (B4MED) of the European Institute of Oncology in Milan. Sample collection by the Biobank, in the presence of patient consent, was approved by the Ethical Committee of the European Institute of Oncology on 6 June 2011, and the samples can be used for retrospective studies without any further approval by the Ethical Committee [27]. The levels of hormone receptors, Her-2 and Ki-67, were ascertained by immunohistochemistry. Breast cancer subtypes were defined as follows: Luminal A-like, ER (estrogen receptor) and/or PgR (progesterone receptor)(+), HER2 (Human epidermal growth factor receptor 2)(−), Ki67 < 20%; luminal B–like: ER and/or PgR(+), HER2(−), Ki67 ≥ 20; triple-negative: ER, PgR, and HER2(−), irrespective of Ki67 score; HER2–positive: HER2(+), irrespective of ER, PgR, or Ki67. ER/PgR positivity was defined as ≥ 1% of immunoreactive neoplastic cells, and HER2 positivity was defined as >10% of neoplastic cells with strong and continuous staining of the cell membrane (3 + by immunohistochemistry) and/or amplified by in situ hybridization techniques, in accordance to the ASCO (American Society of Clinical Oncology)/CAP (College of American Pathologists) guidelines. The samples were selected and evaluated by a trained pathologist. Samples with infiltrating carcinoma were selected to have a tumor cellularity of at least 40%, as assessed by hematoxylin and eosin (H&E) staining. Specimens with in situ carcinoma areas, large necrosis areas and massive flogistic infiltration were discarded. For storage, samples were collected and snap-frozen in liquid nitrogen, frozen in optimal cutting temperature compound, or fixed overnight in 4% formalin and embedded in paraffin. The list of patient samples analyzed in this study is summarized in Appendix A.

### 4.3. Laser Micro-Dissection

For experiments with mouse pancreas, 4 µm thick FFPE sections or 10 µm thick snap-frozen OCT–embedded cryosections of mouse adult pancreas were either processed for protein extraction as whole sections or mounted on polyethylene naphthalate membrane (PEN) slides (Leica No. 11600289) previously UV–photoactivated in a UV crosslinker for 30 min (BLX–254, Bio–Link). FFPE sections were de-paraffinized with two changes of xylene, while OCT–embedded sections were fixed in cold anhydrous ethanol for 3 min before proceeding to partial rehydration in graded alcohols up to 50%. Sections were then counterstained for 30 s with alcoholic-based buffered cresyl-violet freshly prepared (0.8% cresyl violet in 60% EtOH and 4 mM Tris–HCl, pH 8.0), washed twice in 75% EtOH and air dried completely before proceeding to the micro-dissection. Areas of pancreatic tissues corresponding to ~20,000, 5000, 2500, and 1000 acinar cells were micro-dissected using a UV–based LMD7 laser micro-dissection system (Leica Microsystems, Wetzlar, Germany), collected into the caps of 0.5 mL tubes and stored at 4 °C until further processing. Experimental procedures involving animals were performed in accordance with the Italian Laws (D.lgs. 26/2014), which enforces Dir. 2010/63/EU (“Directive 2010/63/EU of the European Parliament and of the Council of 22 September 2010 on the protection of animals used for scientific purposes”). All animal procedures were approved by the OPBA (Organismo per il Benessere e Protezione Animale) of the Cogentech animal facility at the IFOM-IEO Campus, Milan, and authorized by the Italian Ministry of Health. For the experiment shown in Figure 5A,B, 10 µm thick sections from a fresh-frozen breast cancer sample were mounted on PEN slides and stained with hematoxylin. Areas corresponding to normal epithelial cells (four areas, 1800–2100 cells each) or infiltrating carcinoma were collected (two tumor regions, fours areas/tumor region, 4200–4500 cells each) by laser micro-dissection, as described above.

### 4.4. Sample Preparation for MS Analysis

Culture cells (0.5–2*10^6^) were resuspended in 1 mL of phosphate buffered saline (PBS) buffer containing 0.1% Triton X–100 and protease inhibitors. One-tenth of the preparation was lysed by adding 0.1% SDS (whole lysates). Nuclei were isolated from the remaining solution through 10 min centrifugation at 2300× *g*, resuspended in 100 µL of the same buffer containing 0.1% SDS and incubated for a few minutes at 37 °C in the presence of 250 U of benzonase to digest nucleic acids. Protein concentration was evaluated with the bicinchoninic acid assay (BCA, Thermo Fisher Scientific, Waltham, MA, USA), and 5–10 µg of proteins were loaded on a 4–12% precast gel (Thermo Fisher Scientific, Waltham, MA, USA).

Laser micro–dissected tissue pieces were transferred at the bottom of the tubes through 3 min centrifugation at maximum speed and processed through methods previously developed for histone PTM analysis [17], which were adapted to low-abundance samples. FFPE micro-dissected tissue pieces were deparaffinized once in 200 µL hystolemon (Dasit Group Carlo Erba, Cornaredo, Italy) and rehydrated in the same volume of solutions containing decreasing concentrations of ethanol (95%, 50%, 20% ethanol and water). The same rehydration steps were also performed for the frozen micro-dissected samples. Then, all the samples were resuspended in 35–40 µL 20 mM Tris pH 7.4 containing 2% SDS and homogenized by sonication in a Bioruptor device, through 10 cycles (30 s on/30 s off) at high potency. For FFPE samples, proteins were extracted and de-crosslinked at 95 °C for 45 min and 65 °C for 4 h. The whole volume was then loaded on a SDS–PAGE gel. OCT and FFPE whole sections (10 µm thick, which contain approximately 1.5 × 10^6^ cells) were collected in 1.7 mL tubes and processed as described in [17]. Protein content was evaluated with the BCA Protein Assay kit (Thermo Fisher Scientific, Waltham, MA, USA), and 10 µg of protein extract (corresponding to approximately 1/3 of the total preparation) were loaded on a gel. A large band (20–45 kDa) around the size of histone H1 variants was excised for in–gel digestion with trypsin [28].

### 4.5. LC-MS/MS

The MS analysis was performed on a 50 cm EASY-Spray column connected online to a Q Exactive HF instrument through an EASY-Spray™ Ion Source (Thermo Fisher Scientific, Waltham, MA, USA). Solvent A was 0.1% formic acid (FA) in ddH_2_O and solvent B was 80% ACN plus 0.1% FA. Peptides were injected in an aqueous 1% trifluoroacetic acid (TFA) solution at a flow rate of 500 nL/min and were separated with a 95 min 3%–60% gradient of solvent B (80 min 3–30%, 10 min 30–40%, 5 min 40–60%), at a flow rate of 250 nL/min. The Q Exactive HF instrument was operated in the data-dependent acquisition (DDA) mode to automatically switch between full scan MS and MS/MS acquisition. Survey full scan MS spectra (*m*/*z* 375–1650) were analyzed in the Orbitrap detector with a resolution of 60,000 at *m*/*z* 200. The 10 most intense peptide ions with charge states comprised between 2 and 4 were sequentially isolated to a target value for MS1 of 3 × 10^6^ and fragmented by higher-energy collisional dissociation (HCD) with a normalized collision energy setting of 28%. The maximum allowed ion accumulation times were 20 ms for full scans and 80 ms for MS/MS, and the target value for MS/MS was set to 1 × 10^5^. The dynamic exclusion time was set to 20 s, and the standard mass spectrometric conditions for all experiments were as follows: spray voltage of 1.8 kV, no sheath and auxiliary gas flow.

### 4.6. MS Data Analysis

Acquired raw data were analyzed using the integrated MaxQuant software v.1.6.2.3 (Max Planck Institute of Biochemistry, Planegg, Germany [12]), which performed peak list generation and protein identification using the Andromeda search engine. The Uniprot HUMAN_1802 and MOUSE 2019 databases were used for peptide identification. Enzyme specificity was set to trypsin, and two missed cleavages were allowed. Methionine oxidation and *N*-terminal acetylation were included as variable modifications, and the FDR was set to 1%, both at the protein and peptide level. The label-free software MaxLFQ [11] was activated as well as the “match between runs” feature (match from and to, matching time window = 2 min). The LFQ values for histone H1 variants were extracted from the “protein groups” MaxQuant output file and analyzed using Perseus [29] and GraphPad Prism (GraphPad). Changes in histone H1 variant levels between two different conditions were evaluated by two-stage linear step-up procedure of Benjamini, Krieger and Yekutieli, with FDR = 5%, while changes among three conditions were evaluated by one-way ANOVA. iBAQ scores (calculated by dividing a protein’s total intensity by the number of tryptic peptides between 6 and 30 amino acids in length) were also generated by the MaxQuant algorithm and used to evaluate the abundance of a protein relative to the others present in the same sample [30]. The LFQ values for histone H1 variants in the samples described in the manuscript are reported in Appendix A. The mass spectrometry proteomics data have been deposited to the ProteomeXchange Consortium [31] via the PRIDE partner repository with the dataset identifiers PXD020537 and PXD020524.

### 4.7. Immunofluorescence

Quantitative immunofluorescence microscopy was performed to measure H1.0 levels. Ten thousand cells for each condition (control, H1.0 KD and Quisinostat-treated) were plated onto poly-l–lysine–coated coverslips (Thermo Fisher Scientific, Waltham, MA, USA, 354085) for 30 min prior to fixation with 4% paraformaldehyde (Thermo Fisher Scientific, Waltham, MA, USA, 43368) for 15 min. Cells were then washed three times with PBS and permeabilized with 0.5% Triton–X100 in PBS for 5 min. Immunofluorescence staining was performed by first incubating cells for 1 h in blocking buffer (PBS containing 3% BSA and 0.05% Triton X–100), incubating for 1 h with a primary mouse monoclonal H1.0 antibody (Upstate, 05-629, 1:100), washing three times with PBS, and finally incubating for 1 h incubation with a donkey anti-mouse AlexaFluor^®^ 568 (Thermo Fisher Scientific, Waltham, MA, USA, A10037) diluted 1:400 in blocking buffer. Cells were finally washed three times with PBS and mounted onto glass slides with Vectashield containing DAPI (Vector H–1200). Quantification of the fluorescent signal was performed using Metamorph software.

### 4.8. Quantitative RT-PCR

RNA extraction was carried out using the RNeasy Plus Mini Kit (Qiagen, Hilden, Germany, 74134) following the manufacturer’s instructions, and cDNA was generated using the High Capacity cDNA Reverse Transcription Kit (Merck KGaA, Darmstadt, Germany, 4374966). Gene expression levels were analyzed on a CFX96 real-time PCR detection system using SsoAdvanced™ Universal SYBR^®^ Green Supermix (Bio-rad, Hercules, CA, 1725274), CFX manager 3.0 software and the following primers: H1.0 FW: 5′-CTGGCTGCCACGCCCAAGAA-3′, H1.0 RV: 5′-CGGCCCTCTTGGCACTGGCA-3′; PPIA FW: 5′-GTCAACCCCACCGTGTTCTT-3′, PPIA RV: 5′-CTGCTGTCTTTGGGACCTTGT-3′. PPIA encodes for Cyclophilin A and was used as reference housekeeping gene.

## Figures and Tables

**Figure 1 ijms-21-07330-f001:**
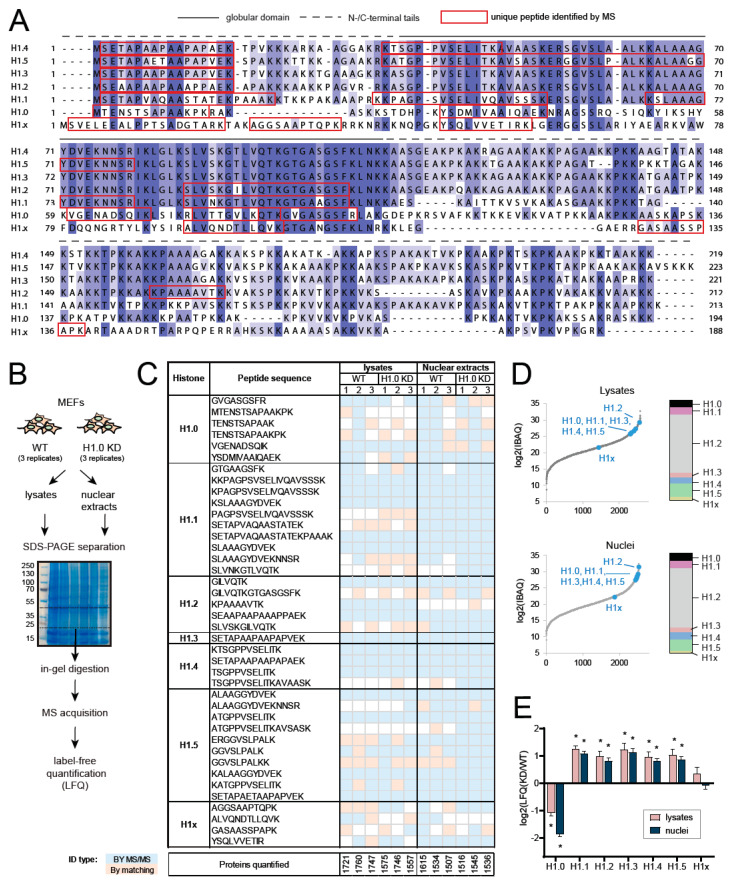
Quantification of histone H1 variants through the MaxLFQ algorithm. (**A**) Sequence alignment of mouse histone H1 variants. Unique sequences detected by MS are marked by red squares. (**B**) Experimental scheme: biological triplicates for protein extracts obtained from mouse embryonic fibroblasts (MEFs) WT, or expressing reduced levels of histone H1 upon shRNA silencing (H1.0 KD), were analyzed by MS. Either lysates or nuclear extracts were loaded on a 4–12% polyacrylamide gel, and a gel band around the size of histone H1 (25–40 kDa) was selected for in-gel digestion with trypsin. (**C**) Unique peptides with up to two miscleavages identified from the samples described in (**B**). Peptides identified in less than three samples are not displayed (refer to Appendix A for the complete list of peptides). The number of proteins identified and quantified from the same samples is also indicated. White spaces indicate that the peptide was not identified and quantified. (**D**) Intensity-based absolute quantification (IBAQ) scores distribution for proteins quantified from the experiment described in (**B**). The position of histone H1 variants along the distribution is shown, and their relative amounts are shown in the bar on the right. (**E**) Histograms showing the log2 of the label-free quantification (LFQ) fold change observed for histone H1 variants in histone H1.0 KD cells relative to WT cells. Error bars represent the standard error of the mean (SEM) from biological triplicates. * *p* value < 0.05 by multiple *t*-test comparison. ID: identification.

**Figure 2 ijms-21-07330-f002:**
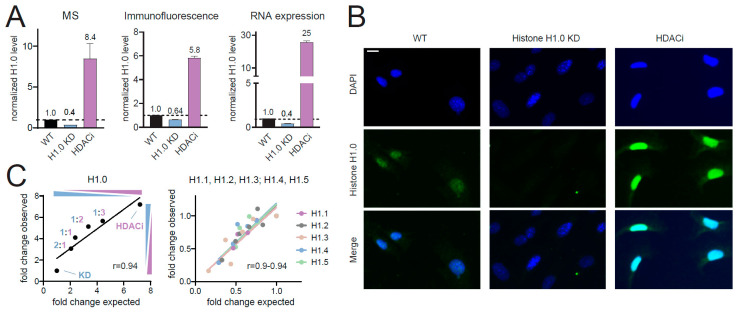
Comparison with other approaches. (**A**) Histone H1.0 protein levels in MEF cells WT, H1.0 KD and treated with the histone deacetylates inhibitor (HDACi) quisinostat, as measured by MS or immunofluorescence (left and central panels), and mRNA levels (right panel) in the same samples. Error bars represent the SEM from biological triplicates. (**B**) Immunofluorescence staining of histone H1.0 in the samples described in (**A**). Scale bar: 10 µm. (**C**) H1.0 KD cells and cells treated with HDACi were mixed in different proportions, and the different histone H1 isoforms were quantified by MS. The plots show the expected and observed fold changes between each sample and the H1.0 KD sample (refer to Appendix A for more details).

**Figure 3 ijms-21-07330-f003:**
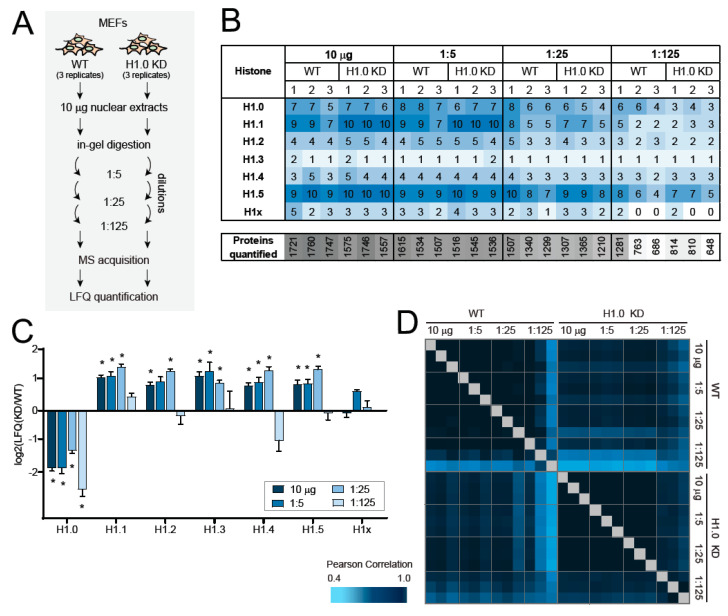
Testing MS–based LFQ quantification of histone H1 variants in low-abundance samples. (**A**) MEFs WT or with lower levels of histone H1 (H1.0 KD) were processed with protocol 2 from Figure 1B. After digestion and prior to MS analysis, the peptides were subjected to 1:5 serial dilutions, starting from 10 µg of proteins loaded on the gel. (**B**) Number of unique peptides identified from the samples described in (**A**). The number of proteins quantified from the same samples is also indicated. (**C**) Histograms showing the log2 of the LFQ fold-change observed for histone H1 variants in histone H1.0 KD cells relative to WT cells. Error bars represent the SEM from biological triplicates. * *p* value < 0.05 by multiple *t*-test comparison. (**D**) Correlation matrix based on Pearson correlation coefficients of log2 MaxLFQ values for histone H1 variant in the nuclear extracts described in (**A**).

**Figure 4 ijms-21-07330-f004:**
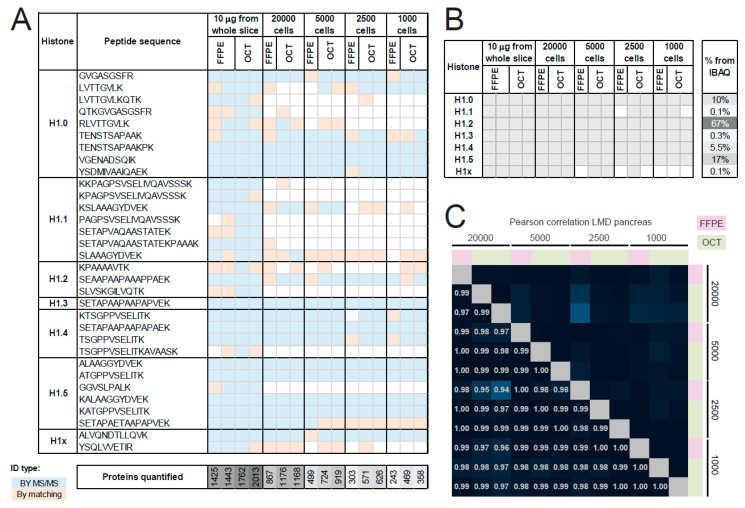
Quantification of histone H1 variants from laser micro-dissected samples. (**A**) Unique peptides with up to 2 miscleavages identified from mouse pancreas whole sections or laser micro-dissected areas (20,000, 5000, 2500 and 1000 cells). Peptides identified in less than three samples are not displayed. At least three samples comprising both formalin-fixed paraffin-embedded (FFPE) and optimal cutting temperature (OCT) tissues were analyzed for each condition. White spaces indicate that the peptide was not identified and quantified. The number of proteins quantified from the same samples is also indicated. (**B**) Histone H1 variants quantified through the MaxLFQ algorithm from the samples described in (**A**). The % abundance of the different variants is indicated on the right. (**C**) Correlation matrix based on Pearson correlation coefficients of log2 MaxLFQ values for histone H1 variant in the laser micro-dissected samples shown in (**A**). ID: identification.

**Figure 5 ijms-21-07330-f005:**
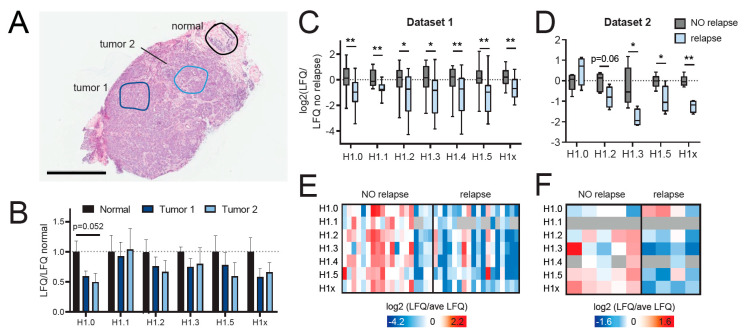
Quantification of histone H1 variants from patient samples. (**A**) Haematoxylin and eosin (H&E) staining of a breast cancer section. Normal epithelial cells and tumor cells (areas containing 1800–4500 cells from four consecutive sections) in the indicated areas were collected by laser micro-dissection (LMD) and analyzed by MS. Scale bar: 2 mm. (**B**) MS analysis of histone H1 variants from the tissue areas highlighted in (**A**). The histograms show the log2 of the LFQ fold-change relative to the average of the normal samples for each histone H1 variant. Error bars represent the standard error of the mean (SEM) from 4 laser micro-dissected areas. Tumor areas were compared to normal areas by one-way ANOVA, followed by Dunnett’s multiple comparison test. (**C**,**D**) Boxplot representation of the H1 variant LFQ values in triple-negative breast cancer samples with or without relapse three years after chemotherapy. The samples in (**C**) were stored in OCT, and whole slices were processed using the procedure described in Figure 1, while the samples in (**D**) were stored as FFPE tissues and whole proteomes where acquired. * *p* value < 0.05, ** *p* value < 0.01 by multiple *t*-test comparison. (**E**,**F**) Heatmap display of histone H1 variant levels for the samples described in (**C**,**D**), respectively. Log2 transformed MaxLFQ values, normalized over the average value across the samples are shown. The grey color indicates those variants that were not quantified.

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
