# Peer review of "Label-Free Mass Spectrometry-Based Quantification of Linker Histone H1 Variants in Clinical Samples"

_ijms, 2020, doi:10.3390/ijms21197330_

Round 1
Reviewer 1 Report
In this research article the authors evaluate a workflow for label-free quantification of protein isoforms. With the workflow at hand they were subsequently able to determine protein abundance of histone H1 isoforms from breast cancer tissue.
Although label-free quantification with MaxQuant (MQ) is a common approach in proteomics studies, this manuscript has a lot of strengths. I detail these are: i) the application of the MQ-workflow to quantify protein isoforms, ii) the evaluation of the suitability of differently treated patient samples for high-throughput scale screening, iii) the determination of sensitivity limits in terms of necessary sample amount and iv) the option to extend the workflow to characterization of relevant PTMs.
There is really not much to improve at this paper. Still, I have two major concerns: First concerns data availability. Here the authors forgot to provide the reviewers login to the datasets, which makes it impossible to judge on the completeness of the dataset. Secondly, the authors should discuss the applicability of spectral library searches in comparison to the match-between-run feature in MQ for detection and quantification of protein isoforms from low amounts of sample (see line 220).
I strongly believe that the current work will be of interest for a broad range of readers. In addition to my major concern some improvements should be made in the manuscript text, which I listed below.
- line 95 and throughout the manuscript: it should be kDa. Please check for spaces between numbers and units.
- line 133: please specify the abbreviation MEF also here in the figure caption as this abbreviation is introduced in the main text after the reference to the figure.
- Figure 1C, line 138 f.. I can’t find the number of proteins identified in this image.
- line 158: Figure 2 B is not fitting the text before the reference. Please refer to the right Figure (2C-D?) and add a reference for figure 2B where appropriate.
- line 170: It is not clear which is the basis for these “expectations”. Please explain.
- Figure 3B: Here it states 2 unique peptides for variant H1.3 in sample WT 1. How can this be as Fig. 1 A shows only one unique peptide in H1.3?
- Line 525: The figure does not match the sentence.
- Figure 5: What is the difference between dataset 1 and 2 whose results are shown in Fig. 5 C-D and 5 E-F? Why there are two datasets?
- Line 328 f. Something with punctuation is wrong here.
- Line 336: CO2
- Line 353: It should be: Samples with infiltrating carcinoma….
- Line 366: pH 8.0 (space is missing)
- Please improve the resolution of images. Especially axis-labeling is very small and hence sometimes blurred.
- Table S1: Please provide a table caption explaining the columns listed.
- Figure S3B: How can the number of peptides for each sequence be higher than 1?
Reviewer 2 Report
The manuscript presesented by Dr. Noberini et al. makes use of label-free mass spectrometry to quantify histone H1 variants. The method was developed on cell lines and murine tissue samples, and later applied to breast cancer patient samples. The method was capable of identifying unique peptide sequences among the different variants. The authors performed a thorough validation of their method by analyzing wild type mouse embyronic fibroblasts (MEF), MEF with histone H1.0 knock-down, and MEF treated with Quisinostat to induce H1.0 production, and comparing their MS results with immunofluorescence quantitation. Their results showed an acceptable correlation between the two orthogonal techniques. Linearity of the MS method was also established. The data was presented clearly.
The following are some questions or comments that I hope could improve the presentation of the study.
- The authors mention that PTMs can modulate histone functions. Were any post-translational modifications such as acetylation, methylation, or phosphorylation considered in this analysis? Does the sample preparation workflow eliminate or preserve these modifications? Were the peptide sequences chosen with possible modifications (or lack thereof) in mind?
- In Figures 1C and 4A, do the white spaces indicate peptides that were not found in the respective samples?
- In Figure 2C, what could account for the deviation from linearity of the fold change observed in H1.0?
- How much tissue material (in terms of mass, cell counts, or comparable measurement) do the whole slice samples in Figure 4 contain? Can a rough estimate be provided?
- In Figure 4A, 2 replicates are shown for each condition(FFPE and OCT). In Figure 4B, correlations are shown for only 1 FFPE. Should there also be 2 replicates here?
- In Line 212, it is mentioned that FFPE tissue samples can have artefactual modifications. What modifications are expected from the FFPE process? Can the modifications be included in the MS search?
- The authors suggest running a reference sample in parallel for MS/MS identification of peptides. In Figure 4A there seems to be several peptides that could not be quantified in the microdissected samples, even by using the "match between runs" feature. Could quantitation of the lower abundance histones still be done effectively when the amount of sample is low?
- In Figure 4B, why are correlations shown only between microdissected samples, but not with whole slice samples? The LFQ values remain high in Figure 4B, but this does not give the whole picture. In Figure 4A we see that there are some peptides that could not be detected in microdissected samples. If accurate quantitation of H1 variants is the goal, then the caveat would be that less abundant variants would fall below the method detection limit when microdissected samples with less than 20,000 cells are used. Please let me know if I am understanding the results correctly here.
- Please check line 328.
